# Vine Tea Extract (VTE) Inhibits High-Fat Diet-Induced Adiposity: Evidence of VTE’s Anti-Obesity Effects In Vitro and In Vivo

**DOI:** 10.3390/ijms252212042

**Published:** 2024-11-09

**Authors:** Wonchul Lim, Seongmin Choi, Jinhak Kim, Kwang-Soo Baek, Minkuk Park, Gakyung Lee, Tae-Gyu Lim

**Affiliations:** 1Department of Food Science & Biotechnology, Carbohydrate Bioproduct Research Center, Sejong University, Seoul 05006, Republic of Korea; wlim@sejong.ac.kr; 2Department of Food Science & Biotechnology, Sejong University, Seoul 05006, Republic of Korea; asp1995@naver.com; 3R&D Division, Daehan Chemtech Co., Ltd., Gwacheon-si 13840, Gyeonggi-do, Republic of Korea; jhkim@dhchemtech.com (J.K.); daehanchemtech@dhchemtech.com (K.-S.B.); 4Department of Integrative Biological Sciences and Industry, Sejong University, Seoul 05006, Republic of Korea; prince0610@naver.com (M.P.); lgg1025@sejong.ac.kr (G.L.); 5Convergence Research Center for Natural Products, Sejong University, Seoul 05006, Republic of Korea

**Keywords:** vine tea extract, high-fat diet, adiposity, lipogenesis, lipolysis

## Abstract

This study focused on evaluating the anti-obesity effects of an extract from *Ampelopsis grossedentata* (Hand.-Mazz.) W. T. Wang, also known as vine tea, in mature adipocytes and high-fat diet-induced obese mice. Vine tea extract (VTE) effectively decreased lipid accumulation in mature adipocytes without cytotoxicity, as confirmed by the regulation of several factors associated with adipogenesis, lipogenesis, or lipolysis. Subsequently, in a 12-week experiment with obese mice, oral VTE administration significantly reduced body weight gain induced with high-fat diet intake. Au-topsy findings showed reduced fat accumulation in various areas without liver damage. The VTE-administered group showed lower serum LDL levels, while increasing HDL, than the high-fat diet-administered group. Analysis of adipose tissue biomarkers indicated VTE’s ability to inhibit adipogenesis and lipogenesis, promote lipolysis, and regulate energy metabolism, contributing to reduced adiposity induced by the consumption of a high-fat diet.

## 1. Introduction

Obesity is a medical condition characterized by excessive accumulation of body fat, which may have negative impacts on health [1]. A body mass index (BMI) of 30 or higher, which is determined by an individual’s height and weight, is typically defined obesity [2]. Obesity increases the risk of various health issues, including heart disease, diabetes, hypertension, and certain cancers [3]. Despite recommendations from the World Health Organization (WHO) and the World Obesity Federation, both established to manage public health, many countries continue to face high rates of overweight and obesity. The primary causes of obesity include poor diet, sedentary lifestyle, genetics, environmental factors, psychological factors, and medical conditions and medications [4].

The development of anti-obesity drugs is an active area of research due to the increasing prevalence of obesity worldwide. Current obesity treatments include medications that act on the central nervous system to suppress appetite, such as phentermine, and those that alter fat absorption, like orlistat. More recent advancements have led to the development of GLP-1 receptor agonists, such as semaglutide (marketed as Ozempic or Wegovy), which have shown significant weight loss effects [5]. However, side effects such as nausea, vomiting, diarrhea, and an increased risk of pancreatitis often accompany these drugs. Long-term safety concerns also persist, particularly regarding cardiovascular and psychiatric effects [6].

Given the side effects and limitations of pharmaceutical treatments for obesity, there is a growing need for alternative approaches, particularly those that leverage natural products [7]. Functional foods and dietary supplements derived from natural sources can offer a safer, more holistic method for obesity prevention. These natural products may include plant extracts, fibers, and other bioactive compounds that promote weight loss or prevent weight gain through various mechanisms, such as enhancing metabolism, reducing fat absorption, or modulating appetite [8]. Developing such products not only addresses the increasing demand for natural and preventative health solutions but also aligns with consumer preferences for safer, non-pharmaceutical interventions.

*Ampelopsis grossedentata* (synonym: *Nekemias grossedentata*) is a plant from the Vitaceae family that is primarily distributed in south-central China and Indochina. It grows in high mountains at an altitude of 800 to 1500 m and is mostly cultivated in Zhangjiajie, a mountainous region in central China’s Hunan province. It is characterized by its thin, oval leaves with pointed tips. In China, it is consumed in the form of teas such as “moyeam” and “vine tea”. About 600 years ago, China’s second tribe, an ethnic minority, made it with their own hands and consumed it. In 1385, the Second Tribe gained recognition for its medicinal properties and presented the king with an offering known as “Tengcha”. It has a unique bitterness and strong aftertaste, and unlike regular green tea, component analysis reveals that it contains components such as flavonoids, polysaccharides, alkaloids, and polyphenols [9]. There are about 45% flavonoids in Xuanji Sapod leaves. More than 35% of these are made up of myricetin, myricitylin, dihydromyricetin, and myricetin-3-O-β-D-galactopyranoside. Research from the past has shown that dihydromyricetin from grape tea extract lowers the activity of pancreatic lipase, stops adipocyte differentiation, improves insulin resistance, and helps with diabetes and obesity. The evidence from experiments using obese rats showed various significant effects, including weight loss, reduction in visceral fat, reduction in abnormal lipid metabolism, and suppression of glucose intolerance [10,11].

In this study, we examined the effectiveness of vine tea extract (VTE) in reducing body fat using a mature adipocyte cell line from mice and obese mouse model fed a high-fat diet. Under in vitro conditions, VTE treatment significantly inhibited preadipocyte differentiation into mature adipocytes, as well as subsequent lipid accumulation in the mature adipocytes. Additionally, oral administration of VTE significantly reduced the high-fat diet-induced weight gain in mice. Biomarkers that help with adipocyte differentiation, adipogenesis, lipolysis, and energy metabolism were effectively controlled by VTE. Therefore, VTE demonstrates a high potential as a functional food material for reducing body fat deposition.

## 2. Results 

### 2.1. VTE Reduces Lipid Accumulation and Glycerol Release in 3T3–L1 Cells

In this study, we evaluate the anti-obesity effect of VTE in murine 3T3–L1 mature adipocytes and diet-induced obese mice. First, to determine whether VTE affects lipid accumulation in 3T3–L1 mature adipocytes, we performed Oil Red O (ORO) staining. Additionally, we investigated the impact of VTE on the viability of 3T3–L1 preadipocytes using an MTS assay. As shown in Figure 1A, the cell viability was decreased by VTE treatment in a dose-dependent manner and showed above 80% of viability by treatment of VTE up to 100 μg/mL compared to the untreated control group. Based on this finding, we examined the extent of lipid changes when 3T3–L1 cells were treated with VTE at noncytotoxic concentrations of 0, 12.5, 25, 50, and 100 μg/mL. Consequently, VTE treatment reduced the Oil Red O dye-stained droplets in a dose-dependent manner (Figure 1B). The degree of glycerol release in mature adipocytes is about 6-fold higher than that in preadipocytes. However, VTE dose-dependently reduced MDI-induced glycerol release (Figure 1C).

### 2.2. VTE Regulates Protein Expression of Adipogenesis, Lipogenesis, and Lipolysis of 3T3–L1 Cells

We performed Western blotting analysis to see how VTE stops lipid buildup in 3T3–L1 cells. Figure 2A shows that VTE decreased the protein expression of PPAR-γ, a transcription factor that plays a role in adipogenesis, and SREBP-1c, a transcription factor that plays a role in lipogenesis. FAS is an important enzyme in de novo lipogenesis because it speeds up the process of changing malonyl-CoA into palmitic acid [12]. ACC is associated with energy metabolism for lipid synthesis and regulates the rate of de novo lipid synthesis. Citrate synthase is a major mitochondrial enzyme that contributes to energy production in the TCA cycle and electron transport chain [13]. These three enzymes have an interconnected relationship in de novo lipogenesis. Citrate synthase generates acetyl-CoA, the initial substrate for fatty acid synthesis, which is then converted by ACC into malonyl-CoA, a key precursor for fatty acid chain elongation. FAS subsequently uses both acetyl-CoA and malonyl-CoA to synthesize long-chain fatty acids, integrating these steps into the de novo lipogenesis pathway [14]. VTE inhibited the expression of FAS and CS, enzymes related to lipogenesis, and promoted phosphorylation of ACC (Figure 2B). Several lipolytic enzymes work in a certain order to break down the neutral lipids that build up inside adipocytes into diacylglycerides and monoglycerides [15]. This ultimately causes triglycerides to be converted into one glycerol and three free fatty acids and released out of the adipocytes. ATGL is known as an enzyme that catalyzes the start of triacylglyceride hydrolysis. Interestingly, VTE not only exhibited inhibitory effects on adipocyte differentiation and lipogenesis but also increased ATGL expression involved in the breakdown of accumulated neutral lipids within adipocytes (Figure 2C). VTE lowered the expression of adiponectin, one of the most important adipokines, and perilipin-1, a lipid-coating protein in adipocytes (Figure 2D).

The above results confirm that several signaling pathways regulate adipocyte differentiation and lipid synthesis. Among them, ERK1/2, a major factor in the MAPK signaling pathway, is known to regulate adipogenesis [16]. VTE treatment significantly inhibited the phosphorylation of ERK1/2 (Figure 2E). The VTE treatment stopped the phosphorylation of CREB, which is part of the process that makes adipocytes differentiate (Figure 2E). The above results suggest that VTE suppresses adipocyte differentiation in 3T3–L1 adipocytes by blocking the MAPK/CREB signaling pathway, which in turn prevents lipid accumulation.

### 2.3. VTE Reduces Body Weight Gain in High-Fat Diet-Induced Mice

Animal experiments verified VTE’s inhibitory effect on fat accumulation within adipocytes. To confirm the impact of VTE intake on fat accumulation induced by a high-fat diet, VTE was orally administered for 12 weeks alongside a high-fat diet. As a result, groups administered VTE at 100 mg/kg B.W. and 200 mg/kg B.W. (VTE-L, VTE-H) showed a significant decrease in weight gain compared to the high-fat diet group. However, there was no significant difference in dietary intake. We calculated dietary efficiency by dividing body weight gain (g/day) by food intake (g/day), taking into account the impact of food intake on weight gain measurements. 

The results indicated that VTE intake reduced dietary efficiency compared to the high-fat diet group, suggesting that VTE intake reduced the rate of weight gain relative to the calorie intake (Figure 3A). We conducted micro-CT imaging to observe the extent of fat accumulation in the mice. The results showed that fat accumulation in the abdominal and adjacent areas was approximately 2.5 times higher in the high-fat diet group compared to the normal diet group, while the VTE intake group showed significantly lower fat accumulation than the high-fat diet group (Figure 3B). Specifically, when examining the weight changes of visceral, subcutaneous, retroperitoneal, and epididymal fat following the bodily incision, we found that VTE significantly reduced the weight of fat tissues in all areas where the high-fat diet had accumulated (Figure 3C). Adding VTE significantly reduced the size of fat cells in the accumulated fat tissues compared to the high-fat diet group (Figure 3D). We speculate that the observed weight loss effect of VTE intake is due to a reduction in body fat accumulation.

### 2.4. Changes Serum Biochemical Parameters in High-Fat Diet-Induced Mice

In several previous studies, the hepatotoxicity and hyperlipidemia in mice induced by a DIO animal model were known [17]. We conducted serum analysis to investigate the hepatotoxicity and dyslipidemia-improving effects of VTE in a high-fat diet-induced obesity animal model. The liver releases AST and ALT into the bloodstream when hepatotoxicity or liver damage occurs, leading to an increase in blood concentration [18]. The oral administration of VTE and orlistat resulted in decreased serum AST and ALT levels, which had increased due to the HFD (Figure 4A). The development of a fatty liver is one of the representative symptoms of high-fat diet-induced obesity. A high-fat diet caused fat accumulation in the liver, which VTE dose-dependently reduced. In particular, the group that consumed a high concentration of VTE showed a similar level of liver fat accumulation as the group treated with the obesity drug orlistat (Figure 4B). We evaluated the expression levels of representative biomarkers related to adipocyte differentiation and lipid accumulation in liver tissue, PPAR-γ, SREBP-1c, and FAS, using Western blotting analysis. As a result, the VTE-treated group decreased the expression of these factors in a way that was similar to what was seen in the morphological analysis of liver tissue (Figure 4C).

We commonly divide cholesterol into two groups: high-density lipoprotein cholesterol (HDL) and low-density lipoprotein cholesterol (LDL). HDL binds to accumulated cholesterol in peripheral blood vessels and transports excess cholesterol to the liver. In contrast, LDL binds to cholesterol in the liver and carries it to the inner wall of peripheral blood vessels [19]. The accumulation of cholesterol on the inner wall of peripheral blood vessels, which leads to cardiovascular disease, underscores the importance of maintaining a balance between HDL and LDL [20]. VTE stopped the drop in HDL and rise in LDL levels in serum caused by HFD to the level seen in the ND group (Figure 4D).

Glucagon-like peptide-1 (GLP-1) is a hormone in the incretin system that plays a crucial role in blood sugar regulation. It promotes insulin secretion and inhibits glucagon release in a glucose-dependent manner, making it known as a treatment for type 2 diabetes [21]. VTE elevated the levels of GLP-1 in the serum in a concentration-dependent manner (Appendix A). This indicates that VTE could aid in blood sugar regulation.

### 2.5. VTE Regulated Protein Expression of Adipogenesis and Lipogenesis in DIO Mice Model 

To elucidate the mechanism of VTE against abnormal body weight gain induced by a high-fat diet, the adipose tissue of mice was used for determining the protein expression of several markers associated with obesity or anti-obesity. As shown in Figure 5A,B, the protein expression of major transcription factors or enzymes including adipogenesis and lipogenesis was upregulated in the HFD group, whereas the VTE group showed a downregulation of the expression of these factors compared to those of the HFD group. VTE raised the levels of lipolysis-related proteins like ATGL, PKA, and perilipin. This caused neutral fats to break down, which in turn led to a decrease in body fat (Figure 5C). Adiponectin is a protein hormone primarily secreted by adipocytes and plays a crucial role in maintaining energy homeostasis, lipid metabolism, and inflammation regulation in the body. The high-fat diet group showed a decrease in adiponectin expression in adipose tissue, whereas VTE significantly increased it. 

In contrast, the orlistat-treated group did not increase the adiponectin expression reduced by the high-fat diet (Figure 5D). Obese individuals typically exhibit high leptin and low adiponectin levels. Phosphorylation of Jun N-terminal kinase (JNK) is associated with adipogenesis [22]. The p-JNK/JNK ratio was increased in the HFD group compared to the ND group and decreased in the VTE and Orlistat treatment groups (Figure 5E). 

### 2.6. Ampelopsin Was Contained in VTE and Has Potential for Anti-Obesity Effects

Vine tea extract is known to have ampelopsin as a bioactive component [23,24]. Studies have already demonstrated the efficacy of this substance in reducing body fat in obese model mice induced by a high-fat diet [25]. HPLC analysis of the components in VTE used in this experiment revealed ampelopsin as the major component (Figure 6). Therefore, the literature suggests that ampelopsin is the functional ingredient responsible for the reduction in body fat associated with VTE. 

## 3. Discussion

A variety of factors, including lack of physical activity, smoking, insufficient sleep, and imbalanced nutrient intake, primarily cause obesity. Among these, the most significant contributor is known to be extreme diets, including high-fat content, sugar, and alcohol [26]. Therefore, this study focuses on identifying materials that can inhibit obesity induced by a high-fat diet. The stored fat in white adipose tissue plays a physiological role in supplying energy to other tissues in need [27]. Excess energy resulting from an imbalance between energy intake and expenditure leads to abnormal fat accumulation, and this excessive buildup of fat is defined as obesity. Key regulators of fat breakdown, such as catecholamines, natriuretic peptides, and growth hormones, initiate lipolysis, which can significantly impact the rate of weight loss [28]. Several studies reported that anti-obesity candidates have lipolytic activity on the MDI-induced glycerol release in 3T3–L1 cells [29,30]. Interestingly, we observed that VTE showed inhibitory effects against glycerol release stimulated by MDI medium in 3T3–L1 mature adipocytes (Figure 1C). These findings can be interpreted as VTE simply reducing the glycerol release induced by MDI treatment. However, it is also meaningful to consider the results from Figure 1B,C together. Figure 1B displays that when VTE was used, the buildup of triglycerides caused by MDI decreased in a way that depended on the dose, with the highest concentration stopping fat buildup by about 80%. Considering this, it can be inferred that the glycerol release in the VTE 100 μg/mL treatment group would be higher compared to the MDI-induced group. These results align with the simultaneous regulation of adipogenesis and lipolysis-related biomarkers by VTE treatment.

As preadipocytes change into mature adipocytes, lipid droplets form inside the cells. This causes adipokines to be released [31]. White adipose tissue predominantly expresses Perilipin-1, one of the several types of Perilipin. Perilipin-1 coats lipid droplets in adipocytes, playing a role in storing cytoplasmic lipids [32]. On the other hand, Perilipin’s role in white adipose tissue is known to inhibit fat breakdown in the absence of PKA stimulation and enhance fat breakdown by approximately 100-fold in the presence of PKA stimulation [33]. VTE enhances perilipin expression, and a high-fat diet reduces PKA, suggesting more effective fat breakdown efficacy. Adiponectin enhances insulin sensitivity and has anti-inflammatory and anti-atherosclerotic effects, contributing positively to metabolic health. In obesity, the blood levels of adiponectin tend to decrease, which is a key factor in increasing the risk of insulin resistance, type 2 diabetes, fatty liver, and cardiovascular diseases [34]. Lower adiponectin levels are known to be associated with a higher likelihood of developing metabolic syndrome-related conditions. Conversely, weight loss, dietary improvements, and exercise can increase adiponectin levels, improving metabolic health and reducing the risk of disease [35].

The deactivation of the ERK1/2 pathway by VTE treatment directly interferes with the blocking of adipogenesis, which means there are fewer chances for fat to build up. IBMX, a commonly used adipocyte differentiation inducer, activates CREB by triggering growth arrest in preadipocytes. Additionally, insulin, continuously administered on differentiation days 1 and 2, activates the MAPK pathway by binding to insulin receptors [36].

In summary, VTE demonstrates anti-obesity effects by reducing lipid accumulation, glycerol release, and regulating protein expression related to adipogenesis, lipogenesis, and lipolysis in 3T3–L1 cells. In high-fat diet-induced mice, VTE intake reduces body weight gain, fat accumulation, and serum biochemical parameters associated with dyslipidemia and hepatotoxicity. Additionally, VTE contains ampelopsin, a bioactive compound known for its anti-obesity properties, which contributes to VTE’s effectiveness in reducing body fat.

## 4. Materials and Methods

### 4.1. Preparation of a Standardized Extract of Vine Tea (VTE) 

The dried leaves of vine tea (*Ampelopsis grossedentata* (Hand.-Mazz.) W. T. Wang) were extracted in 60% ethanol at 60 °C for 3 h, filtered, concentrated, and spray-dried. The extraction process was repeated twice. The HPLC results revealed that the ampelopsin content in VTE was greater than 45%.

### 4.2. Cell Culture, Differentiation, and Treatment

We obtained the 3T3–L1 mouse preadipocytes from the American Type Culture Collection (ATCC CL-173; Manassas, VA, USA). Cells were cultured in high-glucose DMEM supplemented with 10% bovine calf serum (BCS), 100 U/mL penicillin, and 100 μg/mL streptomycin (GIBCO, Waltham, MA, USA) to 100% confluence. We then incubated the cells for an additional two days. The cells were induced to differentiate with MDI medium (DMEM containing 10% fetal bovine serum, 1 μM dexamethasone, 10 μg/mL insulin, and 0.5 mM 3-Isobutyl-1-methylxanthine). After 3 days, the medium was replaced with fresh 10 μg/mL insulin medium and then twice, once every two days with fresh medium and maintained at 37 °C in a humidified 5% CO_2_ atmosphere. The 3T3–L1 cells were treated with the indicated concentrations of VTE during differentiation.

### 4.3. Cell Viability Assay

The cytotoxicity of VTE on 3T3–L1 preadipocytes was assessed by an MTS assay. The 3T3–L1 preadipocytes were seeded into a 96-well plate at a density of 5 × 10^3^ cells per well and cultured in DMEM containing 10% BCS for 24 h. The cells were then treated with 0, 12.5, 25, 50, 100, 200, and 400 μg/mL VTE for 48 h. After 48 h, the MTS + PMS solution was added to each well, and the cells were incubated at 37 °C for 1 h. The absorbance at 490 nm was measured using a Cytation 1 (BioTek, Winooski, VT, USA) at the Biopolymer Research Center for Advanced Materials.

### 4.4. Oil Red O Staining

To determine the effect of VTE on intracellular lipid accumulation, Oil Red O staining was employed. Briefly, we removed the culture medium, washed the cells twice with cold PBS, fixed them in 10% formalin for 10 min, and then stained them with ORO solution for 30 min. The images of lipid droplets were observed and photographed using a microscope (Eclipse E600, Nikon, Tokyo, Japan). To quantify the lipid accumulation, ORO dye in droplets was dissolved in 100% isopropanol, and the absorbance of the supernatant was measured at 490 nm using a Cytation 1.

### 4.5. Glycerol Release

To investigate the lipolysis effect of VTE, the contents of glycerol, the degradation product of triglycerides, were measured. We analyzed the glycerol contents in the culture medium using a Glycerol Assay Kit (Sigma-Aldrich, St. Louis, MO, USA). Briefly, we transferred appropriate glycerol standards and culture medium samples into a 96-well plate. The master reaction mix (a mixture of assay buffer, enzyme mix, ATP, and dye reagent) was added to each of the blank, standard, and culture medium sample wells. The 96-well plate was mixed using a horizontal shaker and incubated for 20 min at room temperature. We measured the absorbance at 570 nm using Cytation 1. Thereafter, the glycerol contents of the culture medium were calculated according to the following equations as concentration of glycerol in sample = absorbance at 570 nm of sample/slope determined from the standard curve.

### 4.6. Western Blotting Analysis

3T3–L1 cells, murine liver, or murine epididymal white adipose tissues (eWAT), were lysed with lysis buffer (20 mM Tris-HCl [pH 7.5], 150 mM NaCl, 1 mM Na_2_ EDTA, 1 mM EGTA, 1% Triton, 2.5 mM sodium pyrophosphate, 1 mM beta-glycerophosphate, 1 mM Na_3_VO_4_, 1 µg/mL leupeptin) (Cell Signaling Technology^®^, Dallas, TX, USA). Protein samples were quantified using the Pierce™ BCA Protein Assay Kit (Thermo Scientific, MA, USA) and applied to sodium dodecyl sulfate-polyacrylamide gel electrophoresis (SDS-PAGE). The separated proteins were transferred from the gel to the polyvinylidene fluoride membrane using Western blotting. The membranes were incubated with the indicated antibodies at 4 °C for 16 h, following the manufacturer’s instructions. A chemiluminescence reader (LuminoGraph 3 Lite; Atto, Tokyo, Japan) was used to detect protein bands.

### 4.7. Animal Experiments

Animal experiments were performed according to the guidelines of Sejong University (approval number: SEMI-23-005, Institutional Animal Care and Use Committee). Four-week-old male C57BL/6 mice were purchased from Orient Bio Inc. (Orient Bio Inc., Seongnam, Republic of Korea) and housed under controlled conditions (24 ± 2 °C, humidity 55 ± 5%, 12 h light/12 h dark cycle) for 12 weeks. After one week of acclimatization, animals were randomly distributed into five groups of 8 animals each, as follows: ND (normal diet, growing rodent diet), HFD (high-fat diet, 60 kcal% fat), VTE-L (HFD + 100 mg/kg of VTE BW/day), VTE-H (HFD + 200 mg/kg of VTE BW/day) and Orlistat (HFD + 20 mg/kg of Orlistat BW/day). Animals were allowed free access to food and water throughout the entire experiment period. Every week, we measured the animal’s food intake and body weight. For twelve weeks, we administered VTE dissolved in sterile normal saline (0.9% NaCl) or Orlistat dissolved in corn oil daily through an oral zonde needle. Animals in the ND and HFD groups were given an equal volume of sterile normal saline by oral zonde needle. After 12 weeks of experiments, the mouse abdomen was incised under respiratory anesthesia with isoflurane (Hanaph, Seoul, Republic of Korea), and blood was collected from the inferior vena cava. The collected whole blood was transferred to a serum-separated tube and incubated at room temperature for 15 min. After that, the tubes were centrifuged at 3000× *g* for 10 min, and serum samples were stored at −80 °C. The liver and white adipose tissues (visceral, subcutaneous, retroperitoneal, and epididymal fat) were obtained, weighed, and stored at −80 °C for further analysis.

### 4.8. In Vivo Micro-CT

To investigate the total fat in mice, in vivo micro-CT (Skyscan 1276, Brucker, Kontich, Belgium) was used to measure the abdominal adipose tissue area and distribution. Micro-CT imaging was performed before autopsy 12 weeks after oral VTE administration. The raw data from the abdomen were acquired at the start and end of the experiment using micro-CT (image pixel size: 39.8 μm, voltage: 70 kV, current: 200 μA, filter: 0.5 mm Al filter, exposure: 140 ms, rotation step: 0.6 deg), and all animals were under respiratory anesthesia with isoflurane to minimize their movement during the scanning. After that, the vertical and transverse abdominal fat area quantifications were calculated using Image J software (National Institutes of Health, Bethesda, MD, USA).

### 4.9. H&E Staining

Liver or epididymal adipose tissues were dissected, fixed in 10% formalin for 24 h, dehydrated with a sequence of ethanol solutions, and processed for embedding in paraffin. Sections of 5 μm in thickness were cut, deparaffinized, rehydrated, stained with hematoxylin and eosin, subjected to microscopic observation, and imaged. Adipocyte morphology was determined using a real-time fluorescence 3D bio image decoding system (Leica Microsystems, Wetzlar, Germany) at the Biopolymer Research Center for Advanced Materials.

### 4.10. Detection of Serum Parameters on Enzyme-Linked Immunosorbent Assay (ELISA)

We measured some serum parameters, including aspartate transferase (AST) and alanine transferase (ALT) for hepatotoxicity. The concentrations of each enzyme were measured using ELISA kits. This experiment was performed according to manufacturer’s protocol for AST or ALT ELISA kits (ab263882, ab282882, Abcam, Cambridge, MA, USA).

### 4.11. Assessment of Serum LDL/VLDL and HDL Cholesterol

To assess dyslipidemia, we measured low-density lipoprotein cholesterol (LDL)/very low-density lipoprotein cholesterol (VLDL) and high-density lipoprotein cholesterol (HDL) levels in serum. This experiment was performed according to the manufacturer’s protocol for a HDL and LDL/VLDL cholesterol assay kit (ab65390, Abcam, Cambridge, MA, USA).

### 4.12. Immunofluorescence

After epididymal adipose tissue sections were deparaffinized and rehydrated, they were fixed and permeabilized with ice-cold methanol for 10 min at −20 °C. The slides were rinsed with PBS three times and blocked using 5% BSA in PBS for 60 min at room temperature. After blocking, peroxisome proliferator-activated receptor gamma (PPAR-γ) primary antibody dissolved in 1% BSA-containing PBS was added to the slides, and the slides were incubated at 4 °C overnight. The supernatant was aspirated, and the slides were washed with PBS five times. Secondary antibodies in PBS were added to the slide and incubated for one hour at 37 °C in a dark condition. The slides were stained with mounting medium with DAPI and covered with coverslips. A fluorescence image was obtained using a real-time fluorescence 3D bio image decoding system (Leica Microsystems, Wetzlar, Germany) at the Biopolymer Research Center for Advanced Materials.

### 4.13. Determination of Ampelopsin in VTE

For standardization of the VTE, HPLC analysis was utilized to determine the phytochemical profile of the VTE with the standard compound, ampelopsin. The analysis used a system equipped with a 2695 Separations Module (WATERS, Milford, MA, USA). The chromatography was established on a Symmetry^®^ C18 (250 × 4.6 mm, 5 μm) column at 27 °C. The elution solvents were 15% acetonitrile and 85% deionized water containing 0.3% phosphoric acid, which was in isocratic mode, and the flow rate was 1 mL/min.

### 4.14. Statistical Analysis

All experiments were performed at least three times, and representative results were presented as the mean and standard deviations. Statistical significance was indicated using Student’s *t*-test analysis or the one-way ANOVA test followed by Tukey’s multiple comparison test. Results were considered significant when the *p*-value was less than 0.05.

## 5. Conclusions

VTE demonstrates anti-obesity effects by reducing lipid accumulation and glycerol release and regulating protein expression related to adipogenesis, lipogenesis, and lipolysis in 3T3–L1 cells. In mice fed a HFD, VTE reduced body weight gain, fat accumulation, and serum biochemical parameters associated with dyslipidemia and hepatotoxicity. In addition, VTE contains ampelopsin, a bioactive compound known for its anti-obesity properties, which contributes to the effectiveness of VTE in reducing body fat.

## Figures and Tables

**Figure 1 ijms-25-12042-f001:**
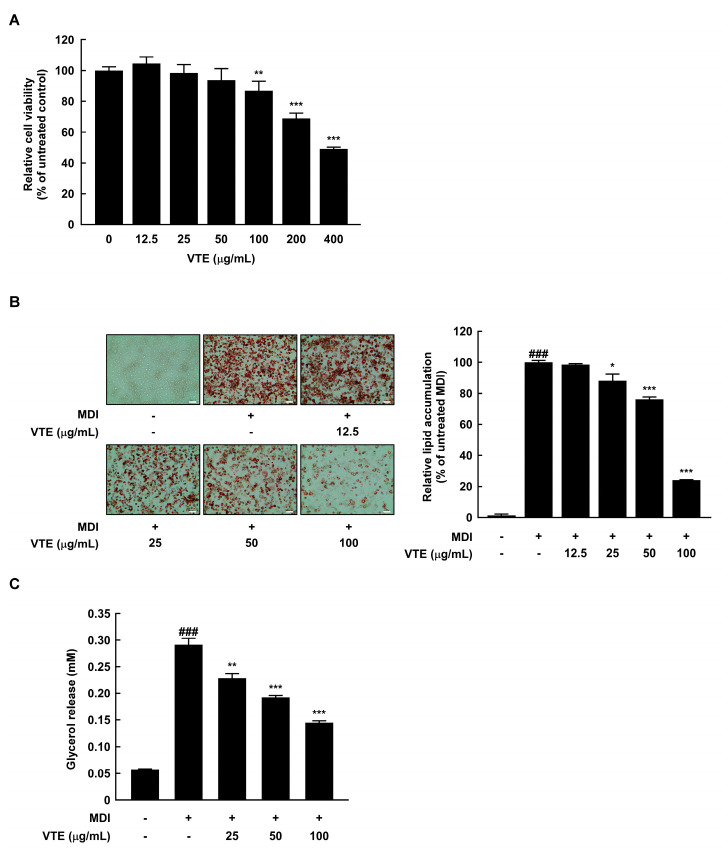
VTE inhibits intracellular lipid accumulation in 3T3–L1 cells. (**A**) 3T3–L1 preadipocytes were treated with VTE (12.5–400 μg/mL) for 48 h and the cell viability was assessed by MTS assay. ** *p* < 0.01, *** *p* < 0.001 as compared to untreated control. (**B**) Intracellular lipid accumulation was visualized using ORO staining and absorbance was measured at 490 nm (white scale bar: 100 μm). To compare lipid accumulation of each group, the degree of cells stained with ORO in control group was expressed as 100%. (**C**) Glycerol release was investigated using a Glycerol assay kit and absorbance was measured at 570 nm using Cytation 1. ^###^ *p* < 0.001 as compared to undifferentiated control. * *p* < 0.05, ** *p* < 0.01, and *** *p* < 0.001 as compared to MDI medium-treated control.

**Figure 2 ijms-25-12042-f002:**
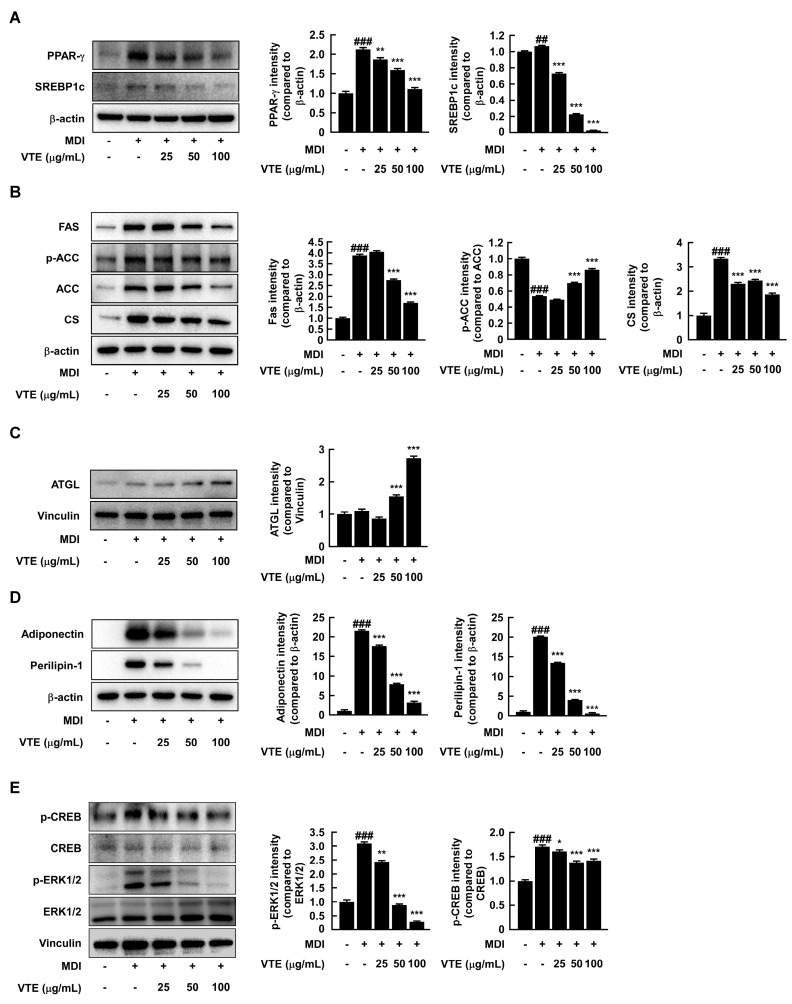
VTE regulates expression of adipocyte proteins. (**A**) Adipogenesis–related protein, (**B**) lipolysis–related protein, (**C**) lipogenesis–related protein, (**D**) Adipokine and modulator of adipocytes, and (**E**) upstream signaling pathway–related protein expression and phosphorylation were assessed using Western blotting analysis. All protein bands were quantified using Image J software. ^##^ *p* < 0.01 and ^###^ *p* < 0.001 as compared to undifferentiated control. * *p* < 0.05, ** *p* < 0.01, *** *p* < 0.001 as compared to MDI medium-treated control.

**Figure 3 ijms-25-12042-f003:**
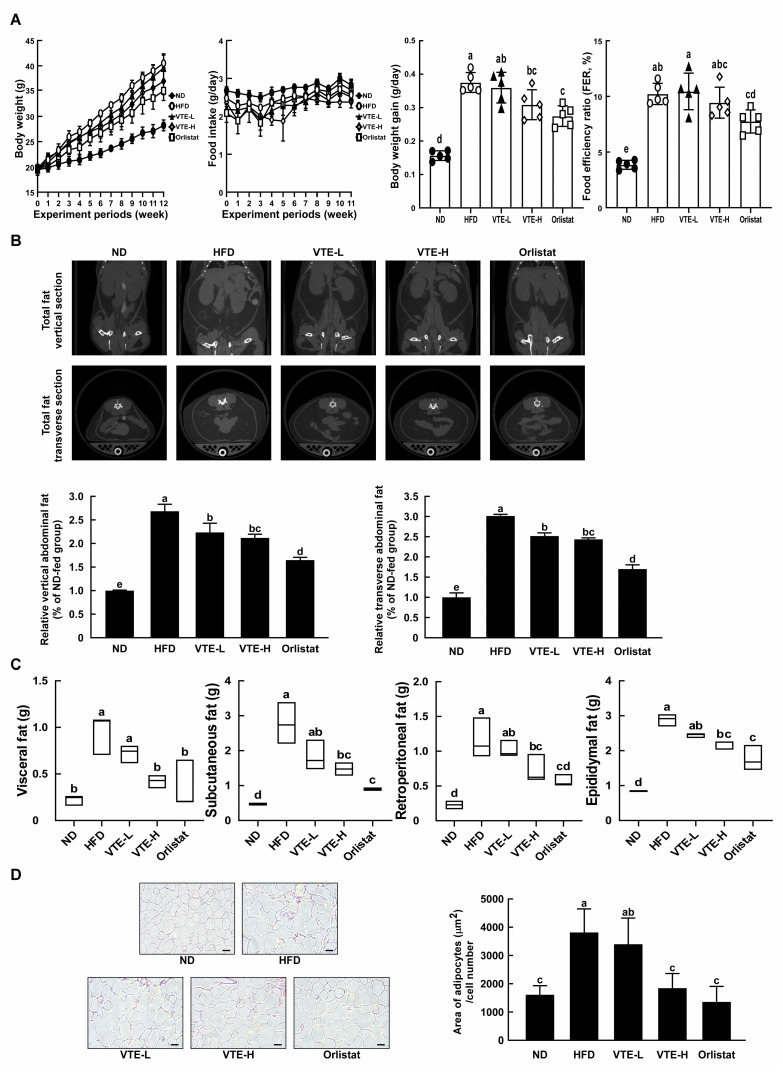
Anti-obesity effects of VTE oral administration in DIO mice model. (**A**) Body weight and food intake were measured for 12 weeks, and body weight gain and food efficiency ratio were calculated as indicated in Material and methods. (**B**) Mice abdominal fat area (vertical and transverse) was measured using micro-CT and quantified by Image J software (version 1.54e). (**C**) After sacrifice, 4 white adipose tissue masses were weighed. (**D**) The size of eWAT was observed by H&E staining and area of adipocytes was calculated by Image J software (version 1.54e) (black scale bar: 100 μm). Statistical analysis was performed using the one-way ANOVA test followed by Tukey’s multiple comparison test. Differential letters (a–d) means statistically significance difference among the groups (*p* < 0.05).

**Figure 4 ijms-25-12042-f004:**
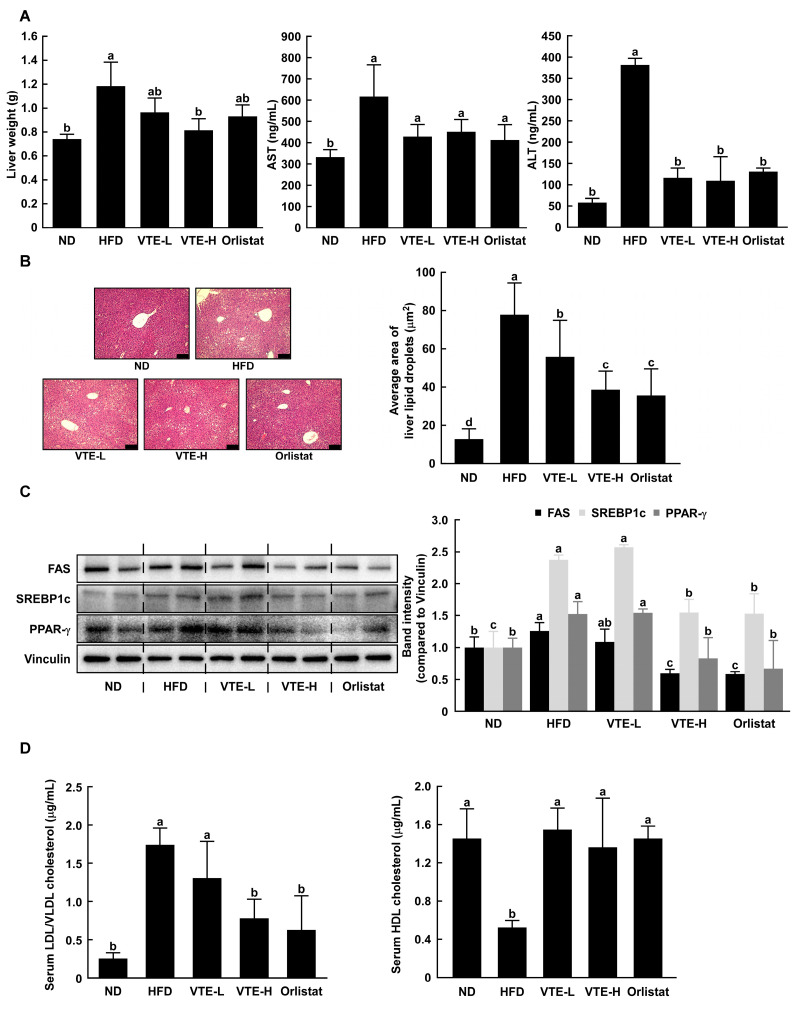
VTE improves hepatotoxicity and metabolic disorder-related biomarkers in mice fed a high-fat diet. (**A**) Murine liver tissue weight was measured after euthanasia. AST and ALT, hepatotoxicity biomarkers, were measured using ELISA kits. (**B**) Area of adipocytes in liver tissue was observed using H&E staining and area of adipocytes were calculated by Image J software (black scale bar: 100 μm). (**C**) In mice liver, adipogenesis and lipogenesis-related protein expression were assessed using Western blotting analysis. (**D**) Indicators of dyslipidemia (serum LDL/VLDL cholesterol and HDL cholesterol) were measured by colorimetric kits. Statistical analysis was performed using the one-way ANOVA test followed by Tukey’s multiple comparison test. Differential letters (a–d) mean statistically significant differences among the groups (*p* < 0.05).

**Figure 5 ijms-25-12042-f005:**
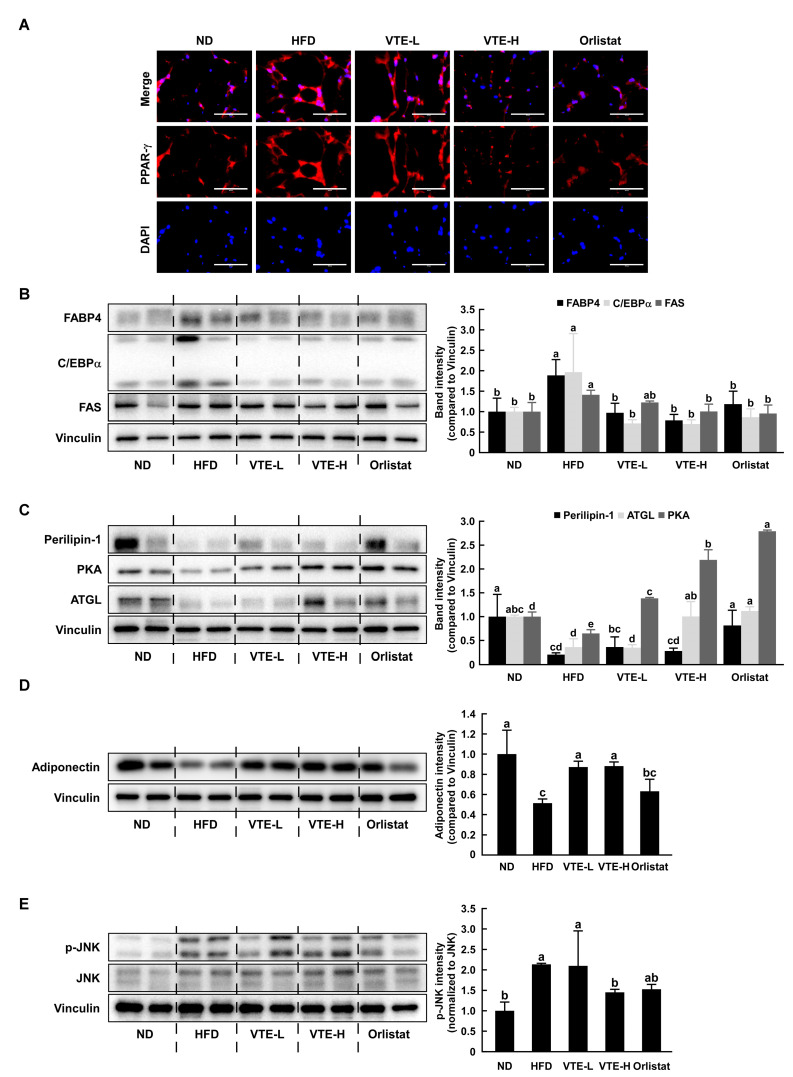
VTE regulates the expression of adipogenesis, lipogenesis, and lipolysis proteins in HFD-induced obese mice. (**A**) PPAR-γ protein expression was evaluated using immunofluorescence (white scale bar: 100 μm). In eWAT of mice, the protein expression related to (**B**) adipogenesis, lipogenesis, (**C**) lipolysis, and (**D**) adipokine was determined using Western blotting analysis. (**E**) Phosphorylation of JNK was determined using Western blotting analysis. All protein bands were quantified using Image J software. Statistical analysis was performed using the one-way ANOVA test followed by Tukey’s multiple comparison test. Differential letters (a–d) mean statistically significant differences among the groups (*p* < 0.05).

**Figure 6 ijms-25-12042-f006:**
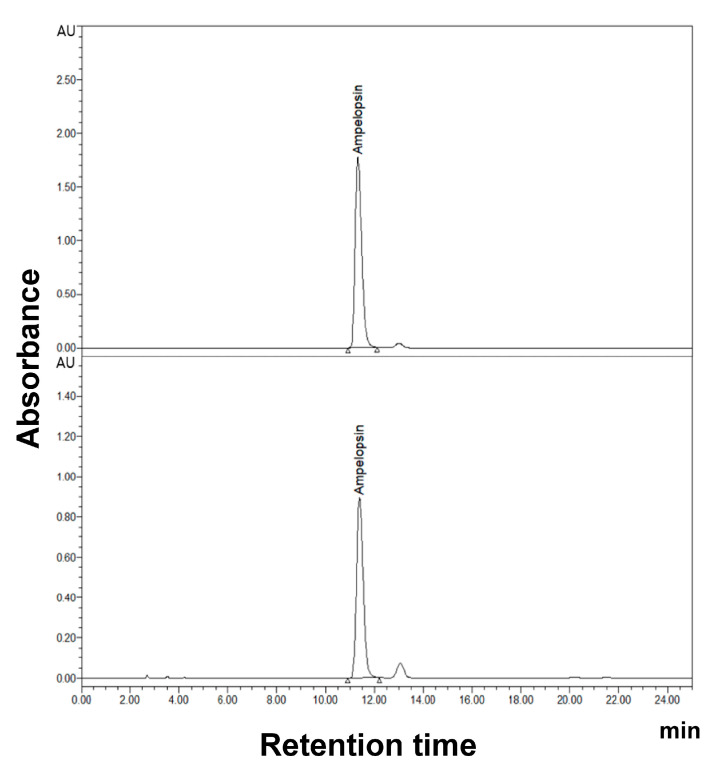
HPLC chromatograms of isolated VTE. Upper is a HPLC chromatogram of ampelopsin standard. Lower is a HPLC chromatogram of ampelopsin, which originated from VTE. Ampelopsin was detected as an active ingredient.

## Data Availability

Data are contained within the article.

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
