# Peer review of "Vine Tea Extract (VTE) Inhibits High-Fat Diet-Induced Adiposity: Evidence of VTE’s Anti-Obesity Effects In Vitro and In Vivo"

_ijms, 2024, doi:10.3390/ijms252212042_

Round 1
Reviewer 1 Report
Comments and Suggestions for Authors
In the present study authors demonstrate the anti-obesity effect of vine tea extract (VTE) in the in vivo model of high fat diet-induced obesity as well as inhibitory effect on lipid accumulation in cultured adipocytes. Consequently, VTE had beneficial effect on glucose and lipid metabolism, adipose tissue accumulation and liver enzyme activities. It is suggested that this natural plant-derived medicine could be considered in the management of overweight and obesity.
The topic is of interest and a lot of data are presented in the paper. However, there are also important concerns to be addressed.
1. The abbreviation “VTE” should be clarified in the title, Abstract and main text when used for the first time.
2. Section 4.4: how ORO was extracted from the cells to be quantified? Was any standard curve generated to calculate stain accumulation?
3. Percent content of proteins and carbohydrates in both diets should be specified. It would also be reasonable to include fat composition (percent of saturated, mono- and polyunsaturated fatty acids) in both diets.
4. In vivo experiments were performed in male mice only; do you expect the same results in female mice?
5. VTE administration was started together with HFD, so the results suggest that VTE partially prevents rather than treats obesity. It would be reasonable to examine therapeutic effect as well, that is to start VTE administration after some time of HFD in mice with already developed obesity.
6. Section 4.8: when (after how many weeks of treatment) microCT was performed?
7. Were ELISA kits used to measure ALT and AST? Was concentration or activity of these enzymes measured?
8. Section 4.14, were all data normally distributed to justify using parametric tests? Was data distribution verified?
9. Lines 95-96: was inhibition of lipolysis the consequence from lower lipid content in the cell or VTE had the independent anti-lipolytic effect?
10. The title of the section 2.6 represents the over-interpretation of the results; it was not demonstrated in this study that amelopsin was responsible for anti-obesity effect of VTE.
11. VTE had no effect on food intake; what mechanism of its anti-obesity activity is suggested? Did it affect food digestion/absorption or energy expenditure in vivo?
Author Response
Reviewer #1
Comment 1. The abbreviation “VTE” should be clarified in the title, Abstract and main text when used for the first time.
Response: We appreciate the reviewer's kind indication. As per the reviewer's comment, all content in the manuscript was changed from “VTE” to “vine tea extract” when used for the first time.
Comment 2. Section 4.4: how ORO was extracted from the cells to be quantified? Was any standard curve generated to calculate stain accumulation?
Response: We appreciate the reviewer's kind indication. To compare lipid accumulation of each group, the degrees of cells stained with ORO in control group was expressed as 100%. In line 100-101, We added the preceding sentence to the figure 1 caption.
Comment 3. Percent content of proteins and carbohydrates in both diets should be specified. It would also be reasonable to include fat composition (percent of saturated, mono- and polyunsaturated fatty acids) in both diets.
Response: We appreciate the reviewer's kind indication. The two diets used in this study are as follows: Normal diet (D10012G; Research Diets Inc, New Brunswick NJ), and 60 kcal% High fat diet (D12492; Research Diets Inc). The components of the two feeds are described in Tables 1 and 2.
Table 1. Normal diet (D10012G).
|
Class description |
Ingredients |
Grams |
|
Protein |
Casein, Lactic, 30 Mesh |
200.00 g |
|
Protein |
Cystine, L |
3.00 g |
|
Carbohydrate |
Starch, Corn |
397.49 g |
|
Carbohydrate |
Lodex 10 |
132.00 g |
|
Carbohydrate |
Sucrose, Fine Granulated |
100.00 g |
|
Fiber |
Solka Floc, FCC200 |
50.00 g |
|
Fat |
Soybean Oil, USP |
70.00 g |
|
Mineral |
S10022G |
35.00 g |
|
Vitamin |
V10037 |
10.00 g |
|
Vitamin |
Choline Bitartrate |
2.50 g |
|
Anti-oxidant |
tert-Butylhydroquinone (tBHQ) |
0.01 g |
Table 2. 60 kcal% high fat diet (D12492).
|
Class description |
Ingredients |
Grams |
|
Protein |
Casein, Lactic, 30 Mesh |
200.00 g |
|
Protein |
Cystine, L |
3.00 g |
|
Carbohydrate |
Lodex 10 |
125.00 g |
|
Carbohydrate |
Sucrose, Fine Granulated |
72.80 g |
|
Fiber |
Solka Floc, FCC200 |
50.00 g |
|
Fat |
Lard |
245.00 g |
|
Fat |
Soybean Oil, USP |
25.00 g |
|
Mineral |
S10026B |
50.00 g |
|
Vitamin |
Choline Bitartrate |
2.00 g |
|
Vitamin |
V10001C |
1.00 g |
|
Dye |
Dye, Blue FD&C #1, Alum. Lake 35-42% |
0.05 g |
Comment 4. In vivo experiments were performed in male mice only; do you expect the same results in female mice?
Response: We appreciate the reviewer's kind indication. According to previous study on 'sex differences in a mouse model of diet-induced obesity: the role of the gut microbiome,' male mice gained more weight than female mice on the same high-fat diet. This suggests that sex-specific differences in the gut microbiota may contribute to variations in weight gain, though scientific evidence is still lacking. Therefore, male mice, which are more sensitive to a high-fat diet, were used in this study. Although not the focus of this study, further research comparing sex differences is necessary.
Comment 5. VTE administration was started together with HFD, so the results suggest that VTE partially prevents rather than treats obesity. It would be reasonable to examine therapeutic effect as well, that is to start VTE administration after some time of HFD in mice with already developed obesity.
Response: We appreciate the reviewer's kind indication. The purpose of this study focused on preventing obesity induced by a high-fat diet. Therefore, VTE administration was started together with HFD.
Comment 6. Section 4.8: when (after how many weeks of treatment) microCT was performed?
Response: We appreciate the reviewer's kind indication. MicroCT imaging was performed before autopsy 12 weeks after oral VTE administration. We added information about timing of microCT scans to lines 378-379.
Comment 7. Were ELISA kits used to measure ALT and AST? Was concentration or activity of these enzymes measured?
Response: We appreciate the reviewer's kind indication. We measured the concentrations of ALT and ALT using ELISA kits as previously described in the manuscript. Their activity was not measured. Based on the reviewer’s comment, we revised the previous sentence in lines 396–397 as follows: “The concentrations of each enzyme were measured using ELISA kits."
Comment 8. Section 4.14, were all data normally distributed to justify using parametric tests? Was data distribution verified?
Response: We appreciate the reviewer's kind indication. We conducted data analysis and statistics using parametric tests, specifically Student's t-test and One-Way ANOVA.
Comment 9. Lines 95-96: was inhibition of lipolysis the consequence from lower lipid content in the cell or VTE had the independent anti-lipolytic effect?
Response: We appreciate the reviewer's kind indication. The glycerol release data in Figure 1C supports that VTE effectively inhibits the differentiation process from preadipocytes to mature adipocytes. The lipolytic effect of VTE is not addressed in Figure 1 but is confirmed in Figure 2C, where an increase in the expression of the lipolytic enzyme ATGL is observed. Therefore, the differentiation into mature adipocytes was inhibited by VTE, leading to less lipid accumulation within the cells. As a result, it appeared as though lipolysis was suppressed.
Comment 10. The title of the section 2.6 represents the over-interpretation of the results; it was not demonstrated in this study that amelopsin was responsible for anti-obesity effect of VTE.
Response: We appreciate the reviewer's kind indication. In accordance with the reviewer’s suggestion, we have revised line 238 to read, " Ampelopsin was Contained in VTE and has Potential for Anti-Obesity Effects."
Comment 11. VTE had no effect on food intake; what mechanism of its anti-obesity activity is suggested? Did it affect food digestion/absorption or energy expenditure in vivo?
Response: We appreciate the reviewer's kind indication. Based on the findings of this study, VTE does not affect food intake but inhibits the differentiation of adipocytes that store energy from a high-fat diet, while also promoting the breakdown of accumulated fat.

Reviewer 2 Report
Comments and Suggestions for Authors
The work presented by Lim et al. refers to a study that evaluates the effects of Vinea Tea Extract – VTE in lipid metabolism using mature adipocytes and high-fat diet-induced obese mice as models. The authors provide substantial evidence for an anti-obesity effect of VTE using both models.
Major comments:
i) Page 1, lines 68-71: “Previous research has shown that the dihy- dromyricetin in vine tea extract stops pancreatic lipase from working, stops adipocyte differentiation, raises insulin resistance, and helps with conditions like diabetes and obesity.” The authors need to explain how stopping pancreatic lipase and raising insulin resistance could be considered beneficial effects that help conditions like diabetes and obesity. Something is wrong.
ii) Page 2, lines 109-110: “FAS is an important enzyme in de novo lipogenesis because it speeds up the process of changing malonyl-CoA into palmitic acid”. This is a rather naïve description of the FAS enzyme and the de novo lipogenesis process. Needs considerable improvement this sentence as well as the following two sentences referring to ACC and Citrate synthase. This metabolic section of the manuscript needs extensive editing of its contents and interconnection.
iii) Page 6, sentence in lines 158-161: The sentence makes no sense. The way its written says that fat accumulation is identical in the high-fat diet group and the VTE intake group, ruling out the protective effect afforded by VTE. The authors need to verify these numbers and make corrections accordingly.
iv) Page 6, lines 185-188. Description of HDL and LDL in the manuscript are rather naïve and need to be changed to more scientific descriptions.
v) Some of the metabolic terminology used by the authors in the manuscript is not the most suitable. Page 9, line 250: “supplying lipid energy”; expression needs correction – energy derived from lipids; Page 9, line 252: “unnecessary buildup”; why the use of the term unnecessary in this context. It could be excessive but unnecessary is something else.
vi) Page 9, sentence in lines 257-259: “These findings can be interpreted as VTE simply reducing the glycerol release induced by MDI treatment.”; the authors should relate data to alterations in metabolic pathways, and reduction in glycerol can in fact be seen as a reduction in lipolytic activity, which seems something that contrasts with a more efficient elimination of lipids afforded by VTE treatment.
Comments on the Quality of English Languagei) Page 1, line 21: “… than the only high-fat diet administered group.” Expression needs correction. Simply remove the word only.
ii) Page 1, lines 29-30: “A body mass index (BMI) of 30 or higher, which is determined by an individual's height and weight, is typically defined.” Sentence needs correction. Ending the sentence by “is typically defined” is not the best expression.
iii) Page 1, lines 32-35: The sentence starting by “Even though ….” needs language editing.
iv) Page 2, sentences in lines 61-64: The two sentences need considerable editing. For example, what do the authors mean by “made and consumed it using their hands”? The junction of the terms “made and consumed” associated with “using their hands” sounds weird.
v) Page 2, line 76: “artificial conditions”; the authors need to define what is meant by artificial in the context presented in the sentence.
vi) Page 2, line 79: “increased body weight”; it should be increase in body weight.
vii) Page 2, sentence in lines 79-81: “Biomarkers … were effectively controlled by VTE”. What do the authors really want to say with this sentence?
viii) Page 2 line 81: VTE is defined as a functional material; is this really the best definition of VTE?
ix) Page 2, sentence in lines 86-88: the way the sentence is written makes no sense, mainly by ending it with “using a MTS assay”. Sentence needs editing.
x) Page 2, line 89: the expression “and showed above about 80%” should not be used in this context. Simply remove the term about from it.
xi) Page 2 lines 91-92: “when VTE was treated with MDI at concentrations of 0, 12.5, 25, 50, and 100 µg/mL”. VTE was treated? Needs rephrasing. As is makes no sense.
xii) Page 3, line 113: “concentration-dependently inhibited”; the authors use throughout the manuscript this expression of concentration-dependent effects; other ways to refer to this concentration effect would be more suitable.
xiii) Page 3, lines 116-117: the sentence “This lets the free fatty acids leave the adipocytes” is hanging in the text and is most certainly not the best expression to refer to the process of lipolysis.
xiv) Page 4, legend of figure, line 124: “adipocyte-related protein”; why not simply adipocyte proteins.
xv) Page 4, line 130: “lipid formation”; do the authors mean lipid synthesis?
xvi) Page 6 lines 213-214: “showed downregulated the expression of these factors”; showed downregulated is not the best way to express the observed phenomena. Make correction.
xvii) Page 10, line 307: “twice every two days”; what is this exactly; once a day?
xviii) Page 11, line 357: “10 kcal% fat” and “60% kcal% fat” mean exactly what?
Author Response
Reviewer #2
Major comments:
Comment 1. Page 1, lines 68-71: “Previous research has shown that the dihy- dromyricetin in vine tea extract stops pancreatic lipase from working, stops adipocyte differentiation, raises insulin resistance, and helps with conditions like diabetes and obesity.” The authors need to explain how stopping pancreatic lipase and raising insulin resistance could be considered beneficial effects that help conditions like diabetes and obesity. Something is wrong.
Response: We appreciate the reviewer's kind indication. Inhibiting pancreatic lipase prevents the breakdown of dietary neutral fat and prevents its absorption into the body, which can help with obesity. Increasing insulin resistance can make diabetes worse. To correct the error in that part, the contents of lines 68-71 were changed to “Research from the past has shown that dihydromyricetin from grape tea extract lowers the activity of pancreatic lipase, stops adipocyte differentiation, improves insulin resistance, and helps with diabetes and obesity.”
Comment 2. Page 2, lines 109-110: “FAS is an important enzyme in de novo lipogenesis because it speeds up the process of changing malonyl-CoA into palmitic acid”. This is a rather naïve description of the FAS enzyme and the de novo lipogenesis process. Needs considerable improvement this sentence as well as the following two sentences referring to ACC and Citrate synthase. This metabolic section of the manuscript needs extensive editing of its contents and interconnection.
Response: We appreciate the reviewer's kind indication. We have added information about the interplay between the three enzymes—Citrate synthase, ACC, and FAS—in de novo lipogenesis to lines 113-118 of the manuscript, and included the relevant references.
Comment 3. Page 6, sentence in lines 158-161: The sentence makes no sense. The way its written says that fat accumulation is identical in the high-fat diet group and the VTE intake group, ruling out the protective effect afforded by VTE. The authors need to verify these numbers and make corrections accordingly.
Response: We appreciate the reviewer's kind indication. After checking for errors in the relevant part, the contents to lines 164-167 of the manuscript were changed to "The results showed that fat accumulation in the abdominal and adjacent areas was approximately 2.5 times higher in the high-fat diet group compared to the normal diet group, while the VTE intake group showed significantly lower fat accumulation than the high-fat diet group.”
Comment 4. Page 6, lines 185-188. Description of HDL and LDL in the manuscript are rather naïve and need to be changed to more scientific descriptions.
Response: We appreciate the reviewer's kind indication. In lines 191-194, the terms “good” and “bad” cholesterol have been removed from the manuscript for a more scientific expression.
Comment 5. Some of the metabolic terminology used by the authors in the manuscript is not the most suitable. Page 9, line 250: “supplying lipid energy”; expression needs correction – energy derived from lipids; Page 9, line 252: “unnecessary buildup”; why the use of the term unnecessary in this context. It could be excessive but unnecessary is something else.
Response: We appreciate the reviewer's kind indication. In lines 254-257 of the manuscript, “lipid” was deleted from the awkward expression “lipid energy” and “unnecessary” was replaced with “excessive.”
Comment 6. Page 9, sentence in lines 257-259: “These findings can be interpreted as VTE simply reducing the glycerol release induced by MDI treatment.”; the authors should relate data to alterations in metabolic pathways, and reduction in glycerol can in fact be seen as a reduction in lipolytic activity, which seems something that contrasts with a more efficient elimination of lipids afforded by VTE treatment.
Response: We appreciate the reviewer's kind indication. In general, a decrease in glycerol release can be interpreted as an inhibition of fat decomposition ability. However, only mature adipocytes, not preadipocytes, can produce and decompose fat and release glycerol extracellularly. In conclusion, the results can be interpreted to support that VTE effectively inhibits adipogenesis.
Comments on the Quality of English Language
Comment 1. Page 1, line 21: “… than the only high-fat diet administered group.” Expression needs correction. Simply remove the word only.
Response: We appreciate the reviewer's kind indication. In line 21, the word “only” was deleted from the manuscript.
Comment 2. Page 1, lines 29-30: “A body mass index (BMI) of 30 or higher, which is determined by an individual's height and weight, is typically defined.” Sentence needs correction. Ending the sentence by “is typically defined” is not the best expression.
Response: We appreciate the reviewer's kind indication. The relevant part in lines 29-30 of the manuscript is “A body mass index (BMI) of 30 or higher, which is determined by an individual's height and weight, is typically defined.” Sentence needs correction. Ending the sentence by “is typically defined obesity” was revised.
Comment 3. Page 1, lines 32-35: The sentence starting by “Even though ….” needs language editing.
Response: We appreciate the reviewer's kind indication. The relevant part in lines 32-34 of the manuscript is “Despite recommendations from the World Health Organization (WHO) and the World Obesity Federation, both established to manage public health, many countries continue to face high rates of overweight and obesity.” was changed.
Comment 4. Page 2, sentences in lines 61-64: The two sentences need considerable editing. For example, what do the authors mean by “made and consumed it using their hands”? The junction of the terms “made and consumed” associated with “using their hands” sounds weird.
Response: We appreciate the reviewer's kind indication. We edited the corresponding sentence in the lines 60-61 text as "About 600 years ago, China's second tribe, an ethnic minority, made it with their own hands and consumed it."
Comment 5. Page 2, line 76: “artificial conditions”; the authors need to define what is meant by artificial in the context presented in the sentence.
Response: We appreciate the reviewer's kind indication. In line 75 of the manuscript, the word “artificial” was changed to “in vitro” in this part.
Comment 6. Page 2, line 79: “increased body weight”; it should be increase in body weight.
Response: We appreciate the reviewer's kind indication. We revised the sentence “Additionally, oral administration of VTE significantly reduced the high-fat diet-induced weight gain in mice.” in lines 77-78 of the manuscript.
Comment 7. Page 2, sentence in lines 79-81: “Biomarkers … were effectively controlled by VTE”. What do the authors really want to say with this sentence?
Response: We appreciate the reviewer's kind indication. In lines 78-79, biomarkers refer to proteins such as PPAR-γ, SREBP1c, FAS, and ATGL involved in adipogenesis, lipogenesis, and lipolysis presented in the manuscript. In conclusion, this sentence expresses that VTE exerts an anti-obesity effect by regulating the proteins mentioned above.
Comment 8. Page 2 line 81: VTE is defined as a functional material; is this really the best definition of VTE?
Response: We appreciate the reviewer's kind indication. In the line 80, the words “a functional material” was changed to “a functional food material”.
Comment 9. Page 2, sentence in lines 86-88: the way the sentence is written makes no sense, mainly by ending it with “using a MTS assay”. Sentence needs editing.
Response: We appreciate the reviewer's kind indication. We edit the sentence in lines 85-88 with “First, to determine whether VTE affects lipid accumulation in 3T3-L1 mature adipocytes, we performed Oil Red O (ORO) staining. Additionally, we investigated the impact of VTE on the viability of 3T3-L1 preadipocytes using an MTS assay.”
Comment 10. Page 2, line 89: the expression “and showed above about 80%” should not be used in this context. Simply remove the term about from it.
Response: We appreciate the reviewer's kind indication. The word “about” was deleted from line 89 of the manuscript.
Comment 11. Page 2 lines 91-92: “when VTE was treated with MDI at concentrations of 0, 12.5, 25, 50, and 100 µg/mL”. VTE was treated? Needs rephrasing. As is makes no sense.
Response: We appreciate the reviewer's kind indication. We replace the sentence in lines 90-92 with “Based on this finding, we examined the extent of lipid changes when 3T3-L1 cells were treated with VTE at noncytotoxic concentrations of 0, 12.5, 25, 50, and 100 μg/mL.”
Comment 12. Page 3, line 113: “concentration-dependently inhibited”; the authors use throughout the manuscript this expression of concentration-dependent effects; other ways to refer to this concentration effect would be more suitable.
Response: We appreciate the reviewer's kind indication. Since concentration dependence could not be confirmed in figure 2B in the text, the phrase of lines 118-119 was deleted and modified to "VTE inhibited the expression of FAS and CS, enzymes related to lipogenesis and promoted phosphorylation of ACC.".
Comment 13. Page 3, lines 116-117: the sentence “This lets the free fatty acids leave the adipocytes” is hanging in the text and is most certainly not the best expression to refer to the process of lipolysis.
Response: We appreciate the reviewer's kind indication. The corresponding sentence in lines 121-123 of the manuscript was revised to "This ultimately causes triglycerides to be converted into one glycerol and three free fatty acids and released out of the adipocytes."
Comment 14. Page 4, legend of figure, line 124: “adipocyte-related protein”; why not simply adipocyte proteins.
Response: We appreciate the reviewer's kind indication. We edited “adipocyte-related protein” to “adipocyte proteins” in line 130.
Comment 15. Page 4, line 130: “lipid formation”; do the authors mean lipid synthesis?
Response: We appreciate the reviewer's kind indication. We edited “lipid formation” to “lipid synthesis” in line 136.
Comment 16. Page 6 lines 213-214: “showed downregulated the expression of these factors”; showed downregulated is not the best way to express the observed phenomena. Make correction.
Response: We appreciate the reviewer's kind indication. In figures 5A and 5B, major transcription factors or enzymes including adipogenesis and lipogenesis (PPAR-γ, FABP4, C/EBPα, and FAS) were downregulated in the VTE administration group compared to the HFD group.
Comment 17. Page 10, line 307: “twice every two days”; what is this exactly; once a day?
Response: We appreciate the reviewer's kind indication. This means that 10% FBS media was changed to the cells twice, once every two days. To avoid confusion, the expression in line 312 was revised to "twice, once every two days."
Comment 18. Page 11, line 357: “10 kcal% fat” and “60% kcal% fat” mean exactly what?
Response: We appreciate the reviewer's kind indication. This means that the percentage of fat in the total calories in the diet is 10% and 60%, respectively.

Reviewer 3 Report
Comments and Suggestions for Authors
The authors investigated the anti-obesity effect of vine tea leaves (VTE) in adipocytes and high fat diet-fed mice. VTE lowered the accumulation of intracellular lipids in adipocyte. Moreover, the expression of adipogenic, lipogenic, and lipolytic proteins was reduced by VTE. VTE lowered body weight gain in high fat diet-fed mice. In addition, VTE administration decreased LDL levels, but increased HDL levels in serums. The results are sound. However.
major component of VTE was ampelopsin that carries anti-obesity property. I expect that the novelty would have been reduced. There are concerns that should be addressed.
1. Title should be changed, because VTE is unknown until Materials and Methods part. “vine tea leaves” should be included in the title.
2. 100 ug/mL VTE should be excluded in all experiments, because cell toxicity was observed in this concentration. Thus, lowered intracellular lipid accumulation was not judged which is anti-obesity effect or cell toxicity effect.
3. In Abstract, “VTE lowered the accumulation of intracellular lipids in adipocytes without cytotoxicity” was stated. However, 100 ug/mL VTE showed significant cell toxicity. Thus, this description should be revised including exclusion of the data of 100 ug/mL VTE.
4. Lipolysis should be also evaluated by HSL phosphorylation. HSL phosphorylation is a key step in the regulation of lipolysis.
5. Major component of VTE was ampelopsin that carries anti-obesity property in Fig. 6. In this manuscript, in the first step, major component of VTE should be investigated. If ampelopsin has been identified, the novelty would have been reduced.
Author Response
Reviewer #3
Comment 1. Title should be changed, because VTE is unknown until Materials and Methods part. “vine tea leaves” should be included in the title.
Response: We appreciate the reviewer's kind indication. The title of the manuscript has been modified to reflect the comment 1.
Comment 2. 100 ug/mL VTE should be excluded in all experiments, because cell toxicity was observed in this concentration. Thus, lowered intracellular lipid accumulation was not judged which is anti-obesity effect or cell toxicity effect.
Response: We appreciate the reviewer's kind indication. According to the OECD GUIDELINE FOR THE TESTING OF CHEMICALS, adopted on September 7, 2009, "A chemical substance is considered cytotoxic if it inhibits cell viability by more than 20%." In this study, treatment with VTE at a concentration of 100 µg/mL resulted in a cell viability of approximately 87% compared to the untreated control, indicating no cytotoxicity at that concentration.
Comment 3. In Abstract, “VTE lowered the accumulation of intracellular lipids in adipocytes without cytotoxicity” was stated. However, 100 ug/mL VTE showed significant cell toxicity. Thus, this description should be revised including exclusion of the data of 100 ug/mL VTE.
Response: We appreciate the reviewer's kind indication. Referring to the response to comment 2, since VTE treatment at a concentration of 100 µg/mL does not cause cell cytotoxicity, data at this concentration can also be presented.
Comment 4. Lipolysis should be also evaluated by HSL phosphorylation. HSL phosphorylation is a key step in the regulation of lipolysis.
Response: We agree with your comment. As your comment, HSL phosphorylation is an important step in lipolysis. Interestingly, several previous studies have shown that HFD-induced changes in HSL expression may also be targeted as an anti-obesity target. The references and our experimental results are attached below.
References
- Calderón, Berniza; Huerta, Lydia; Casado, María; González-Casbas, José; Botella-Carretero, Jose; Martín-Hidalgo, Antonia, Morbid obesity–related changes in the expression of lipid receptors, transporters, and HSL in human sperm. Journal of Assisted Reproduction and Genetics 2019, 36, 1.
- Huo, Yanxiong; Lu, Xuhong; Wang, Xiaoyu; Wang, Xifan; Chen, Lingli; Guo, Huiyuan; Zhang, Ming; Li, Yixuan, Bifidobacterium animalis subsp. lactis A6 Alleviates Obesity Associated with Promoting Mitochondrial Biogenesis and Function of Adipose Tissue in Mice. Molecules 2020, 25, 7.
- Fan, Z.; Chen, X.; Liu, T. et al.Pectin oligosaccharides improved lipid metabolism in white adipose tissue of high-fat diet fed mice. Food Sci Biotechnol 2022, 31, 1197-1205
Comment 5. Major component of VTE was ampelopsin that carries anti-obesity property in Fig. 6. In this manuscript, in the first step, major component of VTE should be investigated. If ampelopsin has been identified, the novelty would have been reduced.
Response: We appreciate the reviewer's kind indication. As a result of analyzing VTE through liquid chromatography, ampelopsin was identified as one of the functional components related to anti-obesity effects. Further studies on novel, yet undiscovered compounds are currently underway.

Round 2
Reviewer 1 Report
Comments and Suggestions for Authors
The manuscript has been revised according to the reviewers' comments. I have no further concerns.
Author Response
Thank you very much for your thoughtful comments and the time you have dedicated to reviewing our manuscript. We are pleased to hear that the revisions have addressed your concerns satisfactorily. Your insightful feedback has been invaluable in enhancing the quality and clarity of our work.
Reviewer 2 Report
Comments and Suggestions for Authors
I am pleased with your corrections in the manuscript related to my comments.
Author Response

(The authors gave the same response as above.)

Reviewer 3 Report
Comments and Suggestions for Authors
My concerns were all met. The manuscript was improved.
Author Response

(The authors gave the same response as above.)
